# Computed Tomography Evaluation of Frozen or Glycerinated *Bradypus variegatus* Cadavers: A Comprehensive View with Emphasis on Anatomical Aspects

**DOI:** 10.3390/ani14030355

**Published:** 2024-01-23

**Authors:** Michel Santos e Cunha, Rodrigo dos Santos Albuquerque, José Gonçalo Monteiro Campos, Francisco Décio de Oliveira Monteiro, Kayan da Cunha Rossy, Thiago da Silva Cardoso, Lucas Santos Carvalho, Luisa Pucci Bueno Borges, Sheyla Farhayldes Souza Domingues, Roberto Thiesen, Roberta Martins Crivelaro Thiesen, Pedro Paulo Maia Teixeira

**Affiliations:** 1Institute of Veterinary Medicine, Pará Federal University, Belém 68740-970, Brazil; michelcap3@hotmail.com (M.S.e.C.); rdsa20@gmail.com (R.d.S.A.); kayancunharossy@gmail.com (K.d.C.R.); thiago.cardoso@castanhal.ufpa.br (T.d.S.C.); lucasfilhozaidan477@gmail.com (L.S.C.); luisa_pucci@hotmail.com (L.P.B.B.); shfarha@ufpa.br (S.F.S.D.); betothiesen@ufpa.br (R.T.); thiesenaroberta.crivelaro@gmail.com (R.M.C.T.); ppaulomt@ufpa.br (P.P.M.T.); 2Centro de Educação Profissional DNA, Ananindeua 67133-240, Brazil; josegoncalo1602@gmail.com; 3Campus Araguatins of the Federal Institute of Education, Science, and Technology of Tocantins (IFTO), Araguatins 77950-000, Brazil

**Keywords:** tomography anatomy, cadaver conservation, image diagnostic, morphology, three-toed sloth, xenarthra

## Abstract

**Simple Summary:**

This article demonstrates that computed tomography combined with the use of contrasts for imaging examinations can provide a general and comprehensive topographic view of the vasculature, structures, and organs in frozen and glycerinated cadavers of *Bradypus variegatus*. The objective is to present an alternative technique to the anatomical study of preserved frozen and glycerinated cadavers using computed tomography. The study concludes by highlighting that computed tomography allowed a general and comprehensive view of the anatomical structures of frozen and glycerinated cadavers of *B. variegatus*, such as the topographic location of bone structures, organs, and vessels, with soft tissues better visualized after intravenous or oral administration of contrast.

**Abstract:**

*Bradypus variegatus* has unique anatomical characteristics, and many of its vascular and digestive tract aspects have yet to be clearly understood. This lack of information makes clinical diagnoses and surgical procedures difficult. The aim of this study was to evaluate the anatomical aspects of frozen and glycerinated corpses of *B. variegatus* using computed tomography (CT), emphasizing vascular and digestive contrast studies. Nine corpses that died during routine hospital were examined via CT in the supine position with scanning in the craniocaudal direction. In frozen cadavers, the contrast was injected into a cephalic vein after thawing and, subsequently, was administered orally. In addition to bone structures, CT allowed the identification of organs, soft tissues, and vascular structures in specimens. Visualization of soft tissues was better after contrast been administered intravenously and orally, even without active vascularization. Furthermore, the surfaces of the organs were highlighted by the glycerination method. With this technique, it was possible to describe part of the vascularization of the brachial, cervical, thoracic, and abdominal regions, in addition to highlighting the esophagus and part of the stomach. CT can be another tool for the evaluation of *B. variegatus* cadavers by anatomists or pathologists, contributing to the identification of anatomical structures.

## 1. Introduction

The use of computed tomography (CT) for morphological studies in animal species makes it possible to deeply evidence structural relationships, allowing the visualization of the structures in layers, especially mineralized tissues, with high definition and three-dimensional delimitation of physiological and pathological irregularities. When performed on preserved cadavers, it may reveal anatomical structures similar to the structures and organs of a live animal [1,2].

CT applied to preserved cadavers is a methodology that can be applied to identify pathological characteristics in morphological studies to improve the pathological understanding of species that have peculiar characteristics [3]. Despite postmortem changes in biological tissues, preservation methods applied to cadavers must ensure the maintenance of tissues with minimal morphological and color changes, allowing the identification of anatomical structures and pathological changes [1,4].

Preservation through freezing and glycerination allows anatomical parts to be used for a long period, preserving the morphology and color as close as possible to their original condition, delaying postmortem changes, and ensuring better aesthetic and morphological results, preserving the three-dimensional spatial anatomy, the principles of related structure and function, and anatomical variations, including pathological changes [5,6,7].

CT has an important routine clinical role due to the possibility of identifying organs and cavities that are difficult to assess and distinguishing different types of tissues and structures [8,9]. Still, the possibility of using oral, urinary, and vascular contrasts will optimize the study of organs and tissues by providing greater delimitation [10,11,12].

Due to a poor understanding of some of the morphology of *B. variegatus*, a tomographic study is necessary. The aim of this study was to evaluate the anatomical aspects of frozen and glycerinated bodies of *B. variegatus* using computed tomography, emphasizing studies of vascular and digestive contrast. 

## 2. Materials and Methods

The present study was conducted following the approval of the Animal Ethics and Welfare Committee of Pará Federal University (Protocol No. 5943220321, Belém, Brazil).

For this study, nine cadavers of *B. variegatus* that died during routine procedures at the Veterinary Hospital of the Federal University of Para (UFPA), including six males (puppies) and three females (juveniles), were used. These specimens were classified as infants or puppies when they did not have the characteristic sexual dimorphism of the coat, or juveniles or young when they did have the characteristic sexual dimorphism of the coat [13,14].

The puppies were preserved using the glycerination technique [15]. The process took place in four stages, with the materials being immersed in different types of solutions and packed in plastic boxes at room temperature. The first step was prefixation with a 4% formaldehyde solution for 24 h. Second, dehydration in 70% ethyl alcohol for a week. The third step was the clarification process with 3% hydrogen peroxide for one week. The fourth and last step was the fixation/drying process, in which a bidistilled glycerine solution was added to ethyl alcohol at 99.5% in a 1:2 ratio, respectively. The juvenile individuals were frozen in a horizontal freezer, with temperatures varying between −10 °C (14 °F) and −2 °C (28.4 °F).

No contrast was applied for CT on the cadavers preserved in glycerin. The frozen specimens were defrosted in running water for approximately 2 h. After defrosting, 10 mL of an iodide-base contrast (Optiray 320) was injected into a cephalic vein using a vein catheter cannulated with a disposable syringe with a venous catheter (average pressure of 400–600 psi—pounds per square inch), followed by a CT examination. In a second moment, 10 min after the first examination, 10 mL of the same contrast was administered orally using an esophageal catheter, and a new CT exam was performed.

All cadavers were placed in dorsal recumbency over the rectangular foam pad of the CT scanner, with the thoracic limbs extended cranially and pelvic limbs extended caudally. CT scans were obtained in the craniocaudal direction by use of a 16-channel helical tomographic scanner, model Syngo CT 2009E (Siemens; Forchheim, Germany). The image techniques were 110 kVp and 130 mA. The scan time was 62.66 s with a delay of 3 s. The field of view (FoV) was 115 × 115 mm.

Scanning was obtained at an axial 5 mm interval with multiplanar reformation (MPR) of 1 mm in the coronal and sagittal planes. The acquired images were sent to autoCAD 2021 version 24.0 software (computer-aided design) for three-dimensional reconstruction in VR (Volume Rendering).

All adopted were based on the Nomina Anatomica Veterinaria (NAV) [16].

## 3. Results

### 3.1. Tomographic Images of the Corpses

On CT scans of the nine preserved corpses, it was possible to appropriately identify mineralized tissues and bony structures, allowing the qualitative and quantitative dimensions of structures such as the skull, mandible, spine, ribs, pelvis, and limbs to be delimited. It was possible to visualize a lighter and a darker and denser coloration of the bone structures in cadavers preserved via freezing or glycerination, respectively, as observed in Figure 1, Figure 2, Figure 3, Figure 4, Figure 5, Figure 6, Figure 7 and Figure 8.

### 3.2. Tomographic Images of the Glycerinated Cadavers

CT examination of corpses preserved via glycerination revealed thoracic and abdominal organs without clear definitions, which did not allow for accurate determination of the anatomical structures, although parameters such as the topographic location of these organs filled with glycerin were established. In the abdomen, the stomach was identified, and in the thorax, a structure suggesting the esophagus was identified, both with irregular borders (Figure 5).

### 3.3. Tomographic Images of the Frozen Cadaver with IV Contrast

In the tomographic images of the corpses preserved via freezing that received IV contrast, filling of vessels identified with enhancement was observed, starting from the right cephalic vein and moving into the brachial vein towards the right axillary vessel and the cranial vena cava. From the cranial vena cava, the contrast penetrated until the beginning of the right external jugular vein, but the contrast did not progress in this vein. Still from the vena cava, the contrast distributed to the left axillary vein, but the progress was soon interrupted. The advance in a cranial direction occurred in the right and left vertebral veins (Figure 6A).

The contrast also penetrated from the cranial vena cava in a caudal direction to the heart, partially showing the structure of the cardiac chambers, such as the right atrium and ventriculus. Subsequently, the contrast penetrated through the pulmonary artery and its branches (Figure 6A), completing the contrast of the whole pulmonary tissue (Figure 6B,C).

**Figure 6 animals-14-00355-f006:**
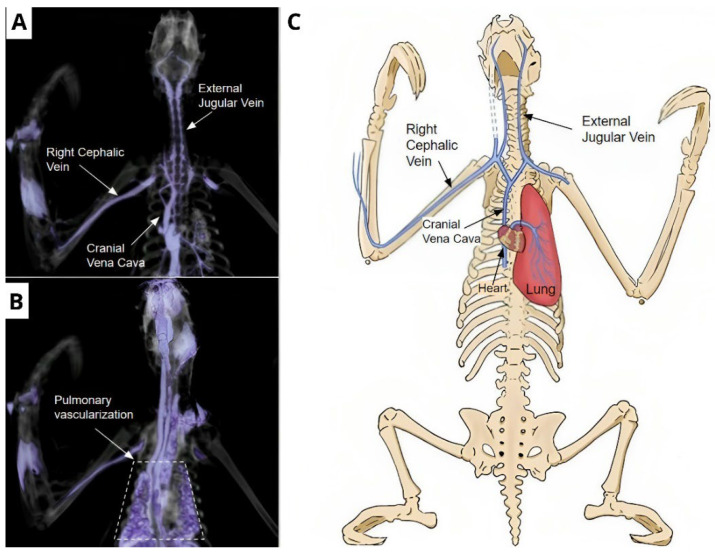
Vascularization of the thoracic region. (**A**) Vascularization starts from the cephalic vein, advancing to the right and left vertebral veins, progresses to the heart until the beginning of the pulmonary artery. (**B**) The contrast penetrated until it fully completed the pulmonary parenchyma. (**C**) Schematic drawing showing the path of the contrast starting in the cephalic vein, enhancing the cardiac and pulmonary tissue.

On the same examination, the contrast progressed through the caudal vena cava, advancing to the hepatic vessels, filling the branches of the hepatic arteries, which were probably a part of the hepatic portal system, and occupying the totality of the hepatic parenchyma (Figure 7A–C).

**Figure 7 animals-14-00355-f007:**
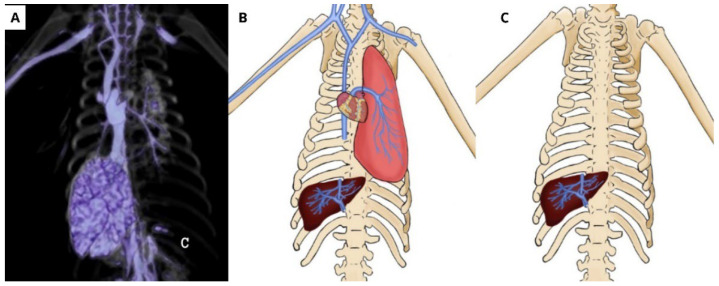
(**A**) Tomographic image showing the contrast progressing through the cranial vena cava, marking the cardiac area, partially showing the right atrium and ventricle, advancing to the vena cava and hepatic veins, fulfilling the whole hepatic parenchyma. (**B**) Schematic drawing showing the path of the contrast through the cranial and caudal. (**C**) Schematic drawing showing the vascularization of the liver.

The contrast enhancement was also visualized in the renal structures using the caudal vena cava, which is ramified, originating from the right and left renal veins, evidencing the kidneys and their location (Figure 8A,B).

**Figure 8 animals-14-00355-f008:**
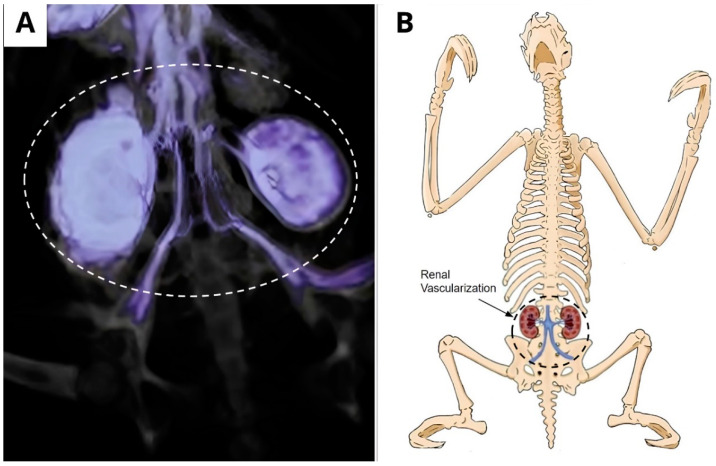
Renal vascularization. (**A**) Tomographic image showing the enhancement of contrast in renal structures by means of the vena cava caudal, which ramified into the right and left renal veins, as well as the disposition of the internal and external iliac veins. (**B**) Schematic drawing presenting renal vascularization demonstrating the kidneys and their location.

It was also possible to visualize the contrast enhancement in the vascular net to perceive branched vessels of the caudal vena cava in its final portion, making it possible to identify the disposition of the external and internal iliac veins in the initial portion of the pelvic limbs, even that the contrast did not continue flowing in the vessel (Figure 9A,B).

**Figure 9 animals-14-00355-f009:**
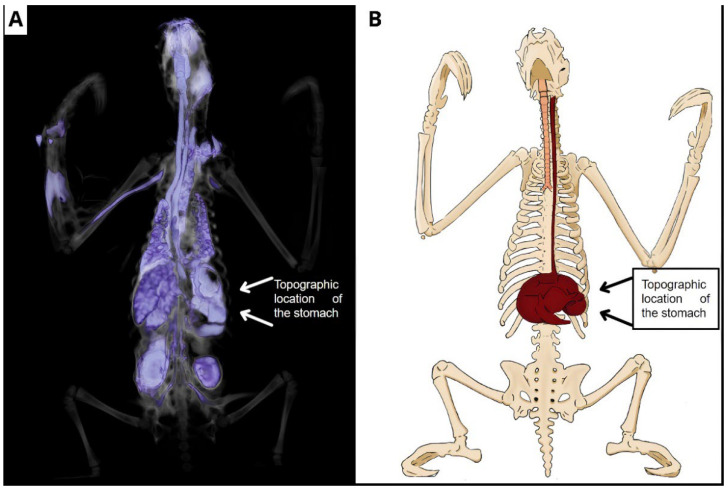
Topographic location of the stomach. (**A**) Tomographic image revealing, after oral contrast, the cranial, thoracic, and abdominal portions of the esophagus, then advancing to the stomach, delimitating its parts. (**B**) Schematic drawing showing the esophagus and stomach, as well as their location.

## 4. Discussion

The bony structures of the frozen specimens showed a lighter coloration, probably due to the preservation of organoleptic characteristics of the corpse during the process, creating images of bony structures very similar to those found in living animals [17]. The darker and denser coloration of the bony structures in the cadavers preserved via glycerination may be related to the formaldehyde used during the glycerination process. This chemical, when associated with glycerin, makes structures opaque, enhancing the level of bone attenuation [18,19].

Computed tomography provides satisfactory results in imaging diagnostics by generating high-quality three-dimensional images that allow a quick and accurate assessment capable of elucidating anatomical variations and pathological changes [20]. However, in the specimens preserved via glycerination, this method created topographic images without definitions and quality delineations of soft tissue, such as those that occur in bony structures, highlighting only the topographic localization of the stomach and part of the esophagus in the thoracic region, as can be seen in Figure 5. The delineation of the visualized structures probably occurred by changing density according to the preservation method [15].

The contrast injected in the right cephalic vein allowed normal flow to the cranial vena cava, reflowing through the vertebral veins due to their smaller diameter. However, the amount of contrast was insufficient to reflow through the external and internal jugular veins [11]. A similar migration of the contrast was described after the authors verified the migration reached by the contrast in the vertebral region [21].

The use of contrasted techniques may be an important adjunct to anatomical studies through imaging diagnostics because they offer more advantages when used in living animals; they present greater dispersion due to the pressure and flow of the active vessels [10,22]. In the present study, contrast solubility contributed to dispersion as well as the administration of a large volume that was ten-fold higher than the dose indicated for use in living animals (2 mL/kg), even without active vascularization or gastrointestinal motility.

Thoracic vascularization was well elucidated by intravenous injection of the contrast, highlighting in the first moment the beginning of pulmonary vascularization and, in the second moment, its entire vascularization. Recently, some reports described CT examinations in latex-injected dog cadavers for anatomic description, delimiting the airways, which contributed to an understanding of bronchial and bronchiolar morphology in dogs by showing didactic and clarifying images. In two models of the canine lower airway, silicone was injected through the trachea until it was visible under the surface of the lung. Subsequently, helical CT acquisition was performed on a ventrally lying specimen, demonstrating the structural organization of the bronchial tree in CT images [23]. However, there are no clarified reports, especially related to the thoracic vascularization of *B. variegatus*, that contribute to better anatomical comprehension of the species.

Functional hepatic vascularization originates specifically from the portal vein. Hepatic vascularization (nutritive) is carried out by the hepatic artery and is drained by the hepatic veins [12]. In the present report, the contrast followed an inverse path, with complete filling of the hepatic parenchyma, allowing for a complete topographic location of the liver in this species [24].

Tomographic studies for the evaluation of renal perfusion are described in laboratory animals to evaluate the damages caused by renal ischemia [25]. Anatomical knowledge of renal vascularization is still in the incipient stages in many species, including *B. variegatus*. Therefore, the present report provides a brief knowledge of renal vascularization in this species, as well as its topographic location, which may be important for understanding the vascular and morphological pathologies of the kidneys in this species.

A renal asymmetry is described in notable anatomical variations within the same species in *B. variegatus*; thus, these findings represent only a preliminary study that requires more research comparing individuals of the same species [26]. The hypothesis that this important finding is due to a disease cannot be excluded [27].

The esophageal rupture in its cervical portion, observed in one of the corpses, is probably related to an attempt to insert an esophageal catheter, as this fact was described in the animal’s medical record, but this alteration may also be due to a bad disposition of the oral contrast or a morphological alteration in the esophagus [28]. The location of the esophagus corroborates the studies that identified the structure and its location in dissected samples of *B. variegatus* [29].

In *B. variegatus*, the stomach is highly complex, presenting four gastric chambers that are subdivided into seven compartments [29,30]. In the exam performed in this study, contrast highlighted only part of these chambers, probably due to the small volume of contrast administered and also to the absence of gastrointestinal motility.

The choice of an image diagnostic for anatomical studies in preserved organisms must take into account the structures or organs to be explored, the preservation method, the protocols and equipment to be used, the operator’s ability, and the techniques and parameters that can contribute to the formation of the images [22,31].

Different ways of preserving cadavers can directly influence the quality of the acquired images, and having this understanding is important, as it allows us to comprehend the limitations of the comparison with the structures in living animals. Even with the difficulty of dispersion of contrast in cadavers, studies in living animals are necessary to elucidate these findings. Imaging diagnostic methods can be another tool used by contemporary anatomists or pathologists for the evaluation of cadavers, either for teaching or research, as they can provide information on the morphology of organs and systems, including pathological changes that may correlate with existing ones in live animals [32].

In general, the tomographic images of the frozen specimens showed a lighter color when compared to the images produced in the glycerinated cadavers, which presented more opaque anatomical structures with attenuation of the bone structures, which were a darker and denser color [18,19]. Methods of preserving corpses can influence the quality of tomographic images [15]. Glycerination can worsen the tomographic image by causing hyperdensity of soft tissues, which is a limitation of the technique in these conditions.

The manuscript provides a general and comprehensive overview of some anatomical aspects, demonstrating that the technique applied to cadavers can provide topographic identification of structures and organs. More studies on a more specific view, directed at the organ or system, need to be carried out. New studies will likely increase the methodology, and the results achieved will be complementary and innovative.

## 5. Conclusions

Computed tomography allowed a general and comprehensive view of the anatomical structures of frozen and glycerinated *Bradypus variegatus* corpses, such as the topographic location of bone structures, organs, and vessels, with soft tissues better visualized after intravenous and oral contrast administration. However, in some tissues, we did not have a complete filling, losing the quality of some images and requiring more details about contrast volumes per live weight and its dissipation.

## Figures and Tables

**Figure 1 animals-14-00355-f001:**
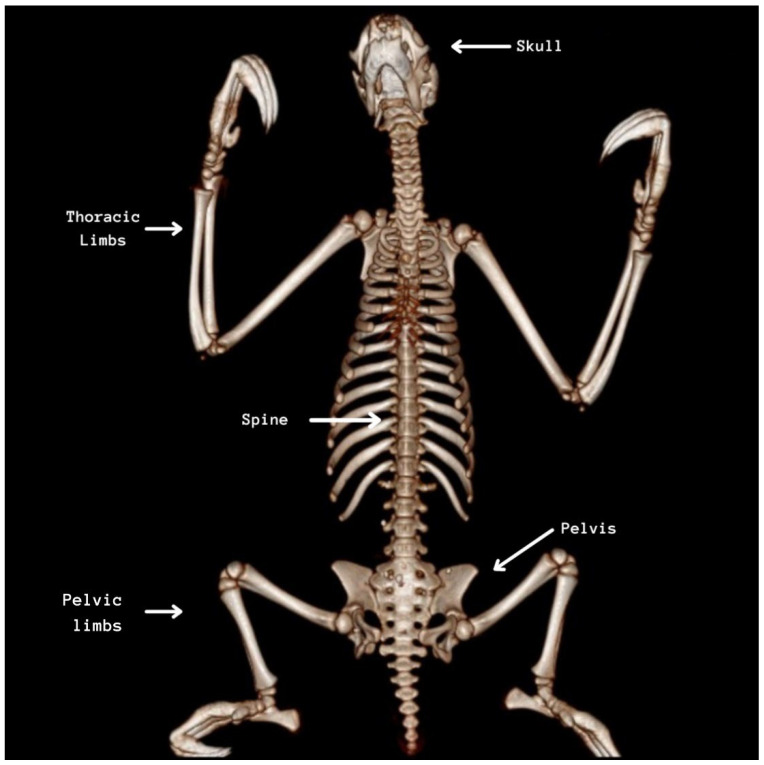
Tomographic image that highlights the bony structures of *Bradypus variegatus*.

**Figure 2 animals-14-00355-f002:**
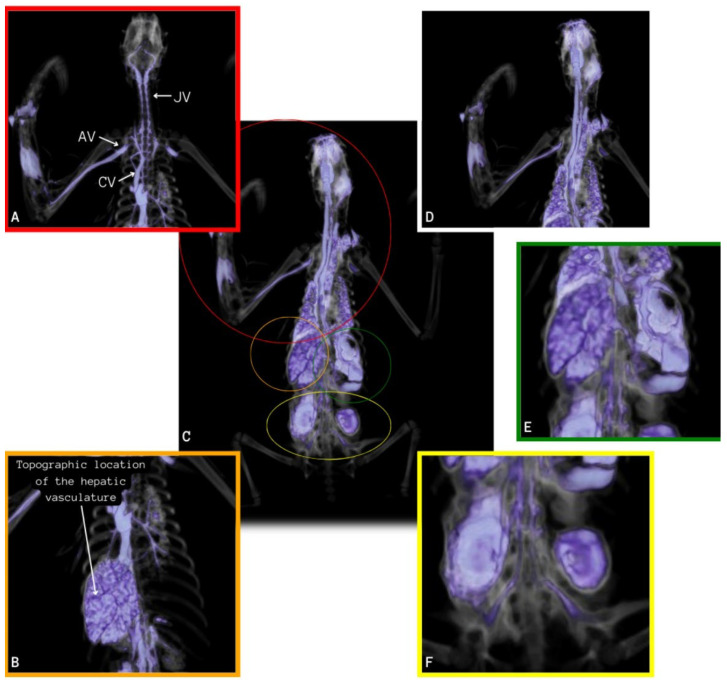
Contrasted image that highlights the soft tissue of *Bradypus variegatus*. (**A**) Topographic location of the right axillary (AV), cranial vena cava (CV), and right external jugular (JV) veins demonstrated by intravenous contrast. (**B**) Topographic location of the hepatic vasculature evidenced by contrast. (**C**) Topographic location of the vascularization of organs and systems evidenced by intravenous contrast. (**D**) Topographic location of the vascularization of organs and systems in the thoracic region evidenced by intravenous contrast. (**E**) Topographic location of organ and system vascularization in the abdominal region evidenced by intravenous contrast. (**F**) Topographic location of the renal vasculature evidenced by intravenous contrast.

**Figure 3 animals-14-00355-f003:**
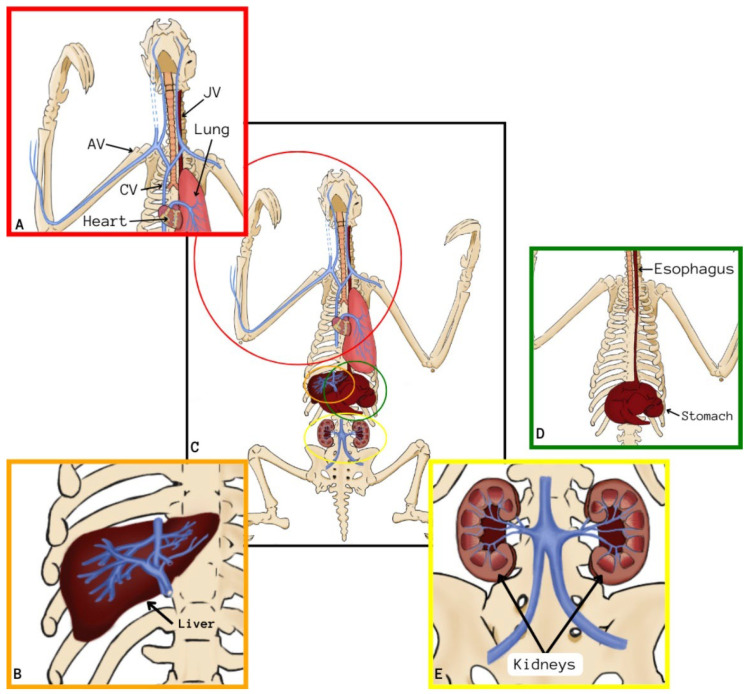
Schematic drawings of the organs and tissues of *Bradypus variegatus*. (**A**) Schematic demonstration of the right cephalic (AV) and axillary veins (AV), cranial vena cava (CV) and right external jugular vein (JV). (**B**) Schematic demonstration of the liver vasculature. (**C**) General view schematically demonstrating the organs and their respective vasculature. (**D**) Schematic demonstration of the topographic location of the esophagus and stomach. (**E**) Schematic demonstration of the renal vasculature.

**Figure 4 animals-14-00355-f004:**
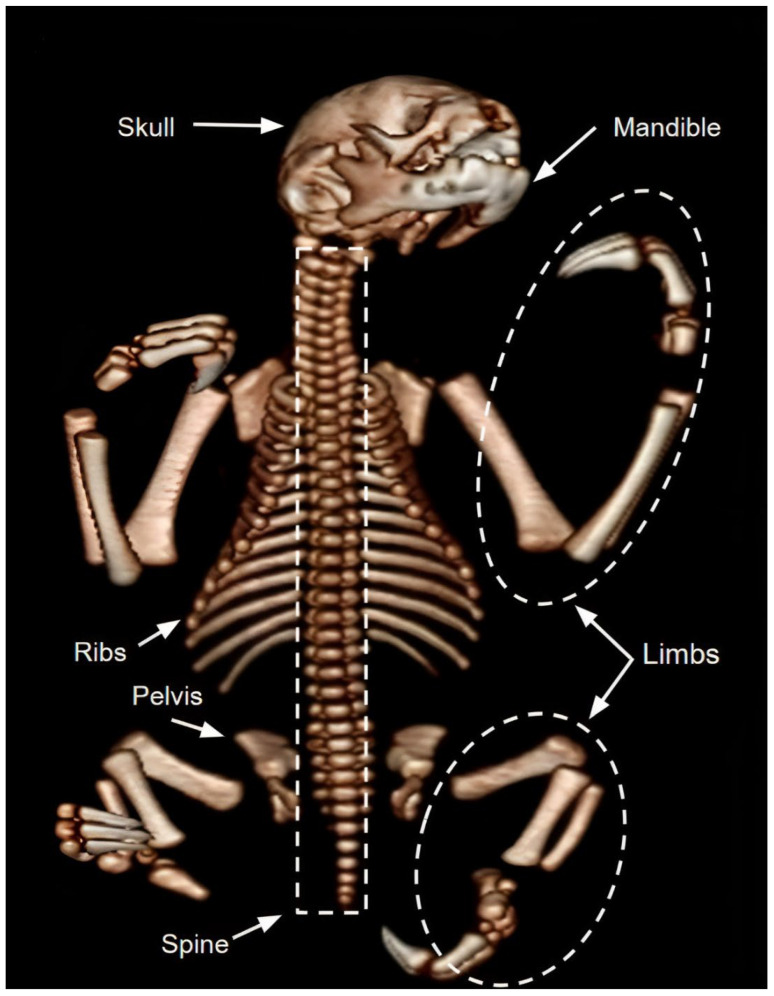
Tomographic images of cadavers from *Bradypus variegatus* preserved in glycerin. Tomographic image enhancing bony tissue.

**Figure 5 animals-14-00355-f005:**
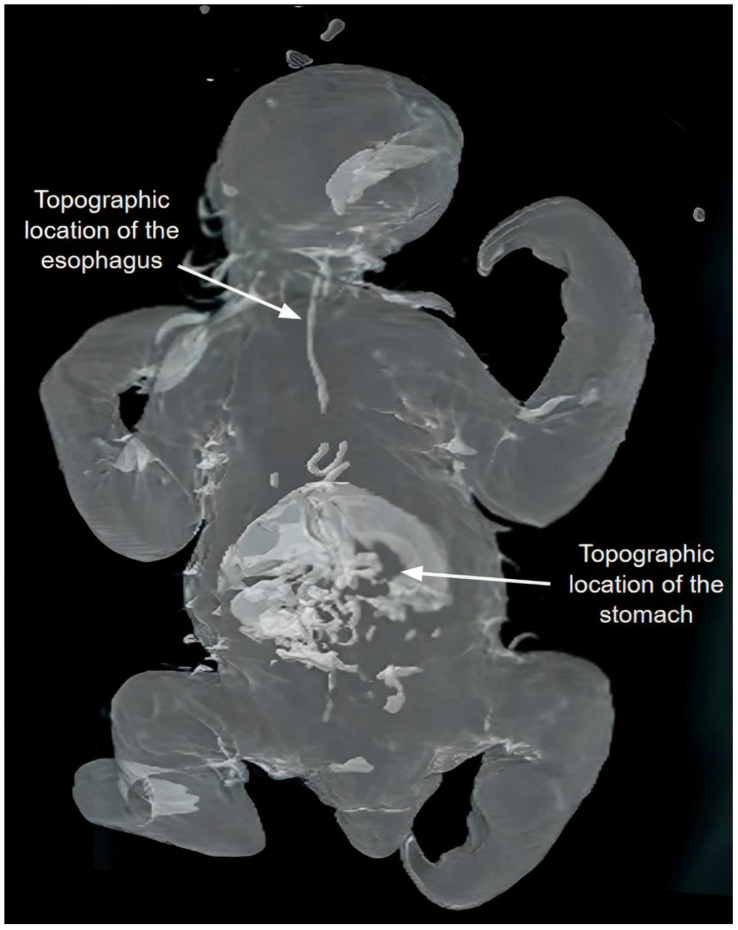
Tomographic image demonstrating the stomach and esophagus contrasted by glycerin.

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
