# Peer review of "Computed Tomography Evaluation of Frozen or Glycerinated Bradypus variegatus Cadavers: A Comprehensive View with Emphasis on Anatomical Aspects"

_animals, 2024, doi:10.3390/ani14030355_

Round 1
Reviewer 1 Report
Comments and Suggestions for Authors
The authors do a good job at conveying what their study is and why it is distinct among previous research and the potential positive outcomes from it. The text all works pretty smoothly, other then a few wording issues here and there. The most important thing that could be improved are the figures. The figures in general are all pretty good, but while I mentioned some improvements to some of them, including labeling some parts and making some larger, all figures could be improved by increasing their sizes. In addition to labeling the different images in the earlier figures, potentially adding labels to some of the features and organs discussed in the text could also help improve some of the schematic figures.

Comments on the Quality of English LanguageOverall very good, just a few minor wording issues, which were noted in the annotated PDF.
Author Response
Dear Reviewer,
Initially, our team would like to thank you for your suggestions and proposed corrections to the manuscript, they were essential for improving the work.
Below are the files with the responses and changes made to the manuscript.
We hope to have answered all your proposals and remain at your disposal.
Best Regards,

Reviewer 2 Report
Comments and Suggestions for Authors. At what pressure was the contrast introduced? Was heparin administered after death?
. It would be convenient to introduce numbering in the figures, marking the structures that have been identified in the CT and in the drawings.
. In post-contrast CT images, although large structures can be seen, it is difficult to distinguish anatomical aspects with precision for detailed study.
Author Response

(The authors gave the same response as above.)

Reviewer 3 Report
Comments and Suggestions for Authors
The anatomical study of Bradypus variegatus using two kind of preservation method in cadavers seems to be a valuable contribution to the comparative anatomy and radiology. The work is well organized, the introduction contains all information needed for reader. The material and methods clearly describes all procedures used in the study. The results are well described and visualised with figures and schemes of very good quality. Discussion is written correctly, due to the lack of wider information about the Bradypus variegatus morphology is must be short. I understand that even that oral administration of contrast was partly interrupted by not predictable problems, the work is a valuable, describing the first attempt of CT use in comparative anatomy. Probably, further studies will increase the methodology and achieved results will be more interesting. I, as an anatomist, would like to read the derailed description of specimen morphology, but I understand that I have to wait for this study continuation. The idea of the work is new and innovative. It use modern imagination methods accessible for veterinary sciences. Today, the interdisciplinary studies can prove that animal anatomy can be not only dead branch of knowledge. I suggest to publish the manuscript after the Native Speaker correction.

Author Response

(The authors gave the same response as above.)

Round 2
Reviewer 2 Report
Comments and Suggestions for Authors
In line 95:
"pelvin lings" should be pelvic limbs
In Fig. 1:
"pelvin limbs" should be pelvic limbs
In Fig.3:
CV does not corresponds with right cephalic. It should be the cranial vena cava (VC)
In the text of the figure, the brachial (BV) is not represented in the image.
Author Response
Dear Reviewer,
We greatly appreciate your contributions, and we hope to have them addressed in the attached manuscript.
We seek to correct all the points raised by you.
We remain at your disposal to make any other corrections and suggestions you deem necessary.
Best Regards,
Appointments:
In line 95:
"pelvin lings" should be pelvic limbs
In Fig. 1:
"pelvin limbs" should be pelvic limbs
In Fig.3:
CV does not corresponds with right cephalic. It should be the cranial vena cava (VC)
In the text of the figure, the brachial (BV) is not represented in the image.
